DATA RELEASE

# Chromosome-level genome assembly of the lemon sole, *Microstomus kitt* (Pleuronectiformes: Pleuronectidae)

Marcel Nebenführ[1,2,3,*,†], David Prochotta[1,2,3,†], Maria A. Nilsson[1,2,3], Menno J. de Jong[1,2,3], Tunca D. Yazici[1,2,3], Fabienne Langefeld[2], Malambo Muloongo[4], Helena Woköck[2], Jakob Jilg[2], Sina C. Bender[2], Marvin M. Zangl[2], Juan-Manuel Ortega Guatame[3,5], Kimberley Williams[2], Moritz Sonnewald[6] and Axel Janke[1,2,3]

1 Senckenberg Biodiversity and Climate Research Centre (BiK-F), Frankfurt am Main, Germany
2 Institute for Ecology, Evolution, and Diversity, Goethe University, Frankfurt am Main, Germany
3 LOEWE-Centre for Translational Biodiversity Genomics (TBG), Frankfurt am Main, Germany
4 Department of Aquaculture & Fisheries Management, University of Ibadan, Nigeria
5 Centre for Wildlife Genetics, Senckenberg Research Institute and Natural History Museum, Gelnhausen, Germany
6 Senckenberg Research Institute, Department of Marine Zoology, Section Ichthyology, Frankfurt am Main, Germany

## ABSTRACT

**Background:** The lemon sole (*Microstomus kitt*) is a culinary fish from the family of righteye flounders (Pleuronectidae), inhabiting sandy, shallow offshore grounds of the North Sea, western Baltic Sea, English Channel, Great Britain and Ireland, Bay of Biscay, and coastal waters of Norway.

**Findings:** Here, we present a chromosome-level genome assembly of the lemon sole. We applied PacBio HiFi sequencing on the PacBio Revio system to generate a highly complete and contiguous reference genome.

The resulting assembly has a contig N50 of 17.2 Mbp and a scaffold N50 of 27.2 Mbp. The total assembly length is 628 Mbp, comprising 24 chromosome-length scaffolds. The identification of 99.7% complete BUSCO genes indicates a high level of assembly completeness.

**Conclusions:** The chromosome-level genome assembly of the lemon sole provides a high-quality reference genome for future population-level genomic analyses of this commercially valuable, edible fish.

Submitted: 20 August 2024

* Corresponding author. E-mail: marcel.nebenfuehr.research@gmx.de

† Contributed equally.

Preprint submitted at https://doi.org/10.1101/2025.04.29.651060

**Subjects** Genetics and Genomics, Marine Biology, Bioinformatics

## DATA DESCRIPTION

### Background information

The lemon sole (*Microstomus kitt*) is a commercially relevant, demersal, bottom-dwelling flatfish species. Found on sandy and gravel substrates on the continental shelf and feeding mainly on polychaetes, it reaches lengths of up to 66 cm (but is rarely larger than 40 cm)

and can live for up to 17 years [1]. Females tend to outnumber males in a population and exhibit a larger size [2].

With an estimated catch of 11,000 tons per year and an estimated value of 36 million Euros, the lemon sole constitutes only a small proportion of the European fishing industry [3]. The lemon sole is exclusively available from wild catches by trawling, as farming methods have not yet been developed. Due to strict EU regulations, the population is stable (IUCN: least concern) but has not been studied for its genomic variation or population genomics [4].

The lemon sole belongs to the Pleuronectidae family of righteye flounders – flatfish in which both eyes move to the right side of the body during development – together with the halibut (*Hippoglossus hippoglossus*). This anatomy makes the lemon sole an interesting species for studying developmental biology [5, 6].

Here, we present the chromosome-level genome assembly of the lemon sole, *Microstomus kitt*, which provides important baseline data for future population genetics analyses – for example, as part of monitoring efforts for this important organism for the fishing industry. The genome assembly was performed over a six-week master's course at Frankfurt Goethe University, Germany [7].

## Sampling, DNA extraction, and sequencing

We obtained a male specimen of the lemon sole (NCBI:txid106175) in July 2023 on an annual monitoring expedition to the Dogger Bank of the North Sea (54°48.271′ N 1°25.077′ E), carried out with permission from the Maritime Policy Unit of the Foreign and Commonwealth Office in the United Kingdom. The sample was frozen at −20 °C and later stored at −80 °C.

We extracted high-molecular-weight DNA from muscle tissue using the QIAGEN DNeasy Blood & Tissue Kit. The quality and quantity of DNA were determined using Qubit Fluorometric Quantification (Thermo Scientific) (RRID:SCR_018095) and the Agilent 2200 TapeStation system (Agilent Technologies) (RRID:SCR_014994).

For long-read libraries, we followed the protocol from the SMRTbell® express template prep kit v.3.0 (Pacific Biosciences – PacBio, Menlo Park, CA, USA) and sequenced at Bioscientia (Ingelheim, Germany) on the PacBio Revio system. We acquired a total of 5.6 million reads, yielding 64 Gbp and a mean read length of 11.5 kbp. To generate long-range data for scaffolding, we also prepared a high-throughput chromosome conformation capture (Hi-C) library. We generated proximity-ligated DNA from heart tissue using the Arima High Coverage Hi-C Kit v.01 (Arima Genomics, Carlsbad, CA, USA) according to the Animal Tissue User Guide. Subsequently, high-coverage Hi-C library preparations were conducted following the protocol from the Swift Biosciences® Accel-NGS® 2S Plus DNA Library Kit, and DNA concentration measured using the Qubit Fluorometer and Qubit™ dsDNA BR (Broad Range) Assay Kit (Thermo Fisher Scientific, Waltham, MA, USA). Quality was assessed through fragment-size distribution using the 2200 TapeStation system (Agilent Technologies). The library was then sequenced on the NovaSeq 6000 platform (RRID:SCR_016387) at Novogene (UK) using a 150 paired-end sequencing strategy, resulting in an output of 269 million raw reads – a total of 40 Gbp of sequencing data.

## Genome size estimation

The genome size of the lemon sole was estimated by analyzing k-mer frequencies. High-fidelity (HiFi) reads were used to determine the k-mer frequency for *K* = 21, running

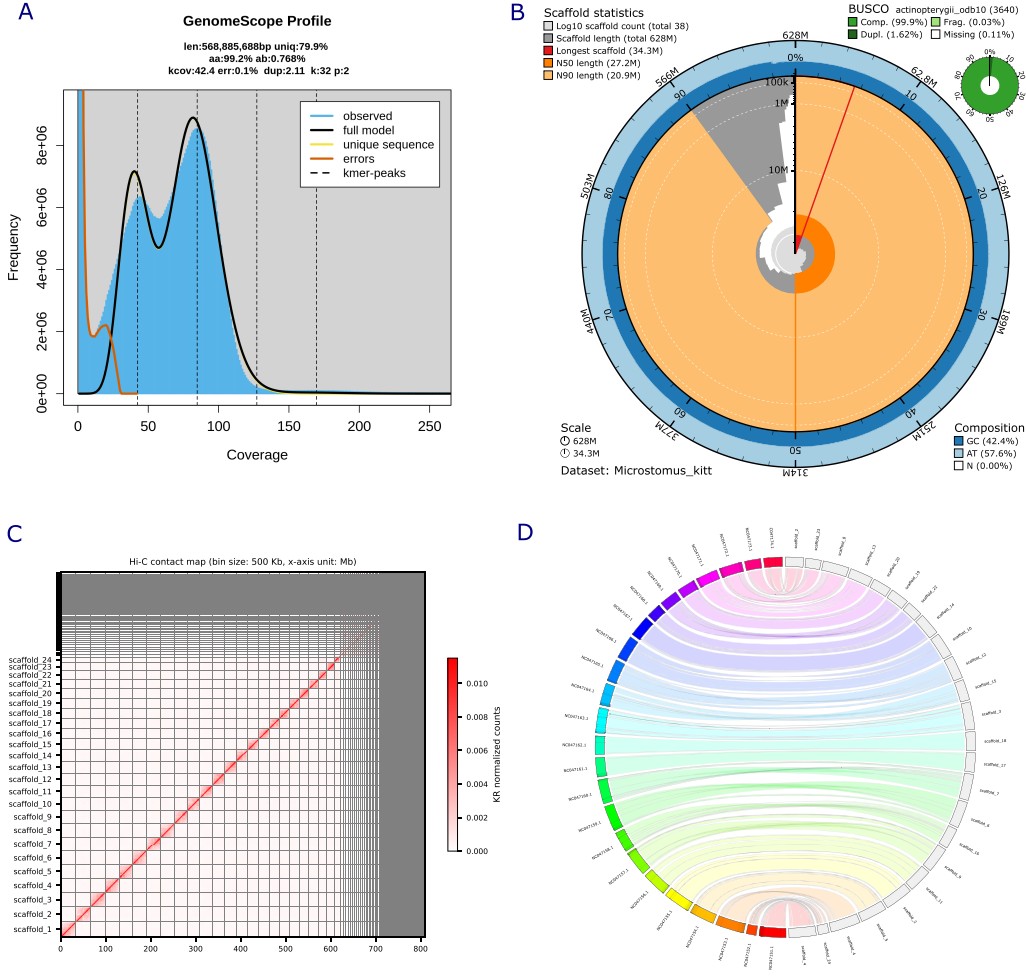

**Figure 1.** **(A)** Estimated k-mer distribution calculated with Jellyfish software and visualized with GenomeScope as a linear plot. **(B)** Snail plot of the *de novo* genome assembly of the lemon sole, showing quality measures. Scaffold statistics are shown by the innermost circle for longest scaffold, N50 and N90 (colors from red to orange). The outer circle (blue) shows GC content, while the green circle shows gene completeness. **(C)** Hi-C contact map of the 24 chromosome-length scaffolds (unplaced contigs unmarked). **(D)** Whole-genome synteny between the final Hi-C scaffolded chromosome-level assembly of the lemon sole (right), and the Atlantic halibut (*Hippoglossus hippoglossus*) genome (GCF_009819705.1; left). Diagonal lines indicate chromosomal rearrangements.

Jellyfish software (Jellyfish, RRID:SCR_005491) v.2.3.0 [8] and GenomeScope v.2.0 [9] (RRID:SCR_017014). The genome size was estimated to be around 542 Mbp, with a heterozygosity of 1.17% (Figure 1A).

## Genome assembly and polishing

We assembled the genome of the lemon sole using Hifiasm (RRID:SCR_021069) v.0.19.5-r587 [10] with standard parameters. The resulting genome was analyzed using Inspector v1.0.1 [11], reporting a total sequence length of 1.09 Gbp long, consisting of 1641 contigs and an N50 of 12.18 Mbp. The mapping rate was 90.15%, and the QV was 38.78. We polished the resulting assembly afterwards three times with Inspector, using default parameters for HiFi reads.

### Assembly QC and scaffolding

To achieve chromosome-length scaffolds, we input the long-read based assembly and the generated Hi-C data into YaHS v1.1 (RRID:SCR:022965) [12] and Chromap v0.2.5 [13]. We mapped the Hi-C reads to the reference genome in Hi-C mode in Chromap, sorted the resulting SAM file, and converted it into the BAM format. After manual curation with JuiceBox v.2.20.00 (RRID:SCR_021172) [14], we have 24 chromosome-level scaffolds. We filtered all scaffolds not containing any BUSCO sequences, as the size and fragmentation suggested they are assembly artifacts. The finalized assembly contained 38 scaffolds (24 chromosome-level) with a total length of 628 Mbp, a scaffold N50 of 27.21 Mbp, and a L50 of 11 (Figure 1B). Subsequently, the genome was gap-filled with TGS GapCloser v.1.2.1 [15]. We evaluated both the raw assembly and the final assembly with Merqury v1.3 (RRID:SCR_022964) in combination with Meryl v1.4.1 (RRID:SCR_026366) [16] and Inspector v1.0.1. We calculated assembly contiguity statistics of the final genome using QUAST v5.0.2 (RRID:SCR_001228) [17] and performed a gene-set completeness analysis using compleasm v.0.2.6 (RRID:SCR_026370) [18] and the provided Actinopterygii orthologous genes database (actinopterygii_odb10).

The final genome was assembled to a total length of 628 Mbp with a scaffold N50 of 27.21 Mbp and a scaffold L50 of 11. The gene-set completeness analysis identified 99.89% complete Benchmarking Universal Single-Copy Orthologs (BUSCO; RRID:SCR_015008) genes (98.22% complete, single copy) and only 0.08% missing BUSCOs, which suggests that the assembly contains most coding regions of the genome (Figure 1B). The Merqury analysis of the raw assembly resulted in an assembly completeness of 90% and a quality value of 60.1, while the final assembly resulted in an assembly completeness of 89.4% and quality value of 56; these decreases could be explained by the scaffolding and gap-filling. We also checked the full 1.1-Gbp sequence, for which a completeness of 90% was reported. The Inspector analysis of the final genome showed a mapping rate of 97.79 % and a depth of 97.04x. The Inspector QV was calculated to be 39.1 (Table S1).

Finally, the Hi-C contact map was created with HapHiC v1.0.7 [19] using the initial Hi-C BAM file from Chromap, and the final assembly graph (AGP), showing 24 chromosome-length scaffolds (Figure 1C).

Genome synteny of the final assembly was visualized using Jupiterplot v1.1 [20] with the Atlantic halibut assembly as reference (*Hippoglossus hippoglossus*; GCF_009819705.1) [21] (Figure 1D). In both genomes, the Jupiterplot analysis was limited to the 24 chromosome-level scaffolds.

Ultimately, the size of the scaffolded and annotated assembly is in the range of other genomes from Pleuronectidae species, and the number of scaffolds is also consistent with the haploid number of chromosome-length scaffolds found in assemblies of other Pleuronectidae species [21, 22].

## GENOME ANNOTATION

### Repeat annotation

To annotate repetitive regions of the lemon sole genome, we used Earl Grey v6.0.1 [23] which incorporates RepeatModeler (RepeatModeler, RRID:SCR_015027) v2.1 [24], and RepeatMasker (RepeatMasker, RRID:SCR_012954) v4.1.6 [25]. Earl Grey was run with species set to Pleuronectidae using RepBase RepeatMasker Edition [26] and DFam v3.7 [27]. In total, 27.61% of the genome consists of repeats (Table 1). After the repeat analysis we



**Table 1.** Repeat content of the lemon sole genome assembly. Repeat content summary with TE ckass statistics found in the genome assembly.

| TE classification | Coverage (bp) | Copy number | % Genome coverage | Genome size | TE family count |
|---|---|---|---|---|---|
| DNA | 59,758,564 | 256,592 | 9.509771377 | 628,391,174 | 822 |
| Rolling circle | 5,929,389 | 34,674 | 0.9435824762 | 628,391,174 | 38 |
| Penelope | 745,793 | 1,597 | 0.1186829209 | 628,391,174 | 11 |
| LINE | 34,470,206 | 108,384 | 5.48546947 | 628,391,174 | 331 |
| SINE | 3,162,509 | 21,212 | 0.5032707541 | 628,391,174 | 25 |
| LTR | 14,962,171 | 44,206 | 2.381028191 | 628,391,174 | 145 |
| Other (simple repeat, microsatellite, RNA) | 21,705,113 | 337,912 | 3.45407668 | 628,391,174 | 10,044 |
| Unclassified | 32,814,865 | 214,530 | 5.222044223 | 628,391,174 | 629 |
| Non-repeat | 454,842,564 | NA | 72.38207391 | 628,391,174 | NA |

**Table 2.** Available flat fish assemblies, used as reference for genome annotation with GeMoMa.

| Species | Accession number |
|---|---|
| Common dab (*Limanda limanda*) | GCF_963576545.1 |
| European flounder (*Platichthys flesus*) | GCF_949316205.1 |
| European plaice (*Pleuronectes platessa*) | GCF_947347685.1 |
| Turbot (*Scophthalmus maximus*) | GCF_022379125.1 |
| Common sole (*Solea solea*) | GCF_958295425.1 |

**Table 3.** Repeat content of the lemon sole genome assembly. Class, class of the repetitive regions. Count, number of occuences of the repetitive region. bpMasked, number of base pairs masked; %masked, percentage of base pairs masked. LINE, Long Interspersed Nuclear Elements (include retroposons); LTR, Long Terminal Repeat elements (including retroposons); SINE, Short Interspersed Nuclear Elements; RC, Rolling Circle.

| Class | Count | bpMasked | %masked |
|---|---|---|---|
| SINEs | 7,761 | 1,314,724 | 0.21 |
| LINEs | 46,143 | 15,290,340 | 2.42 |
| LTR | 21,625 | 8,019,398 | 1.27 |
| DNA transposons | 189,186 | 41,081,369 | 6.49 |
| Rolling-circles | 31,682 | 6,298,121 | 0.99 |
| Unclassified | 342,375 | 70,926,533 | 11.2 |
| Small RNA | 5,630 | 1,825,015 | 0.29 |
| Satellites | 453 | 973,310 | 0.15 |
| Simple repeats | 312,762 | 16,909,154 | 2.67 |
| Low complexity | 31,128 | 1,779,135 | 0.28 |

hardmasked only complex, interspersed repeats (option: **-nolow**) from the resulting repeat library with RepeatMasker.

## Gene annotation

Gene annotation was performed on the hardmasked assembly using GeMoMa v1.9 (RRID:SCR_017646) [28] with high-quality annotated genomes of other flatfish genomes as reference (Table 2). The annotation resulted in 22,713 genes, and 42,982 transcripts (Table 3). Gene-set completeness analysis of the final gene annotation based on the actinopterygii_odb10 database, identified 98.5% complete BUSCOs (62.99% single-copy, 33.79% duplicated), 1.07% fragmented BUSCOs, and 2.14% missing BUSCOs, suggesting high annotation completeness.

## CONCLUSION

Here, we report a chromosome-level genome assembly of the lemon sole (*Microstomus kitt*), a species in the order Pleuronectiformes which is found on sandy bottoms of the northeastern Atlantic Ocean. The annotated genome of the lemon sole, with its high continuity, provides important reference data for future population genetic analyses of this organism.

## DATA AVAILABILITY

All raw data generated in this study – including PacBio HiFi long reads, Hi-C reads, and the chromosome-level assembly – are accessible at GenBank under BioProject PRJNA1153561. Annotation, results files, and other data are available in the GigaDB repository [29].

## ABBREVIATIONS

BUSCO: Benchmarking Universal Single-Copy Orthologs; Hi-C: high-throughput chromosome conformation capture; HiFi: high-fidelity.

## DECLARATIONS

### Ethics approval

The study complied with the 'Nagoya Protocol on Access to Genetic Resources and the Fair and Equitable Sharing of Benefits Arising from Their Utilization.'

### Competing interests

The authors declare that they have no competing interests.

### Authors' contributions

DP, MN, MAN, and AJ designed the study. DP, MAN, MN, AJ, JMO, JJ, MMZ, FL, KW, HW, SCB, performed laboratory procedures and sequencing. DP, MAN, MN, AJ, JMO, JJ, MMZ, FL, KW, HW, SCB, and MdJ conducted bioinformatic processing and analyses. All authors contributed to writing this manuscript.

### Funding

We thank Alexander Ben Hamadou and Charlotte Gerheim from the TBG laboratory centre for laboratory support. The present study is a result of the Centre for Translational Biodiversity Genomics (LOEWE-TBG) and was supported through the programme 'LOEWE – Landes-Offensive zur Entwicklung Wissenschaftlich-ökonomischer Exzellenz' of Hesse's Ministry of Higher Education, Research, and the Arts.

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
