## [Editor Report]

Editor’s AssessmentThis Data Release paper presents the first genome assembly of the lemon sole (Microstomus kitt), a commercially important flatfish found in European coastal waters. It is also interesting that this work was carried out in a University bioinformatics course setting involving students. The resulting chromosome-level genome was assembled using long-read PacBio HiFi sequencing and the Hi-C technique. The 628 Mbp reference (which is consistent with other Pleuronectidae fish species) is assembled into 24 chromosome-length scaffolds with high completeness, achieving a scaffold N50 of 27.2 Mbp. Peer review and data curation made the author clarify a few points and share all of the data and results in an open and well curated manner. The annotated genome of the lemon sole, with its high continuity, should therefore provide important reference data for future population genetic analyses and conservation strategies of this organism.Editor’s AssessmentThis Data Release paper presents the first genome assembly of the lemon sole (Microstomus kitt), a commercially important flatfish found in European coastal waters. It is also interesting that this work was carried out in a University bioinformatics course setting involving students. The resulting chromosome-level genome was assembled using long-read PacBio HiFi sequencing and the Hi-C technique. The 628 Mbp reference (which is consistent with other Pleuronectidae fish species) is assembled into 24 chromosome-length scaffolds with high completeness, achieving a scaffold N50 of 27.2 Mbp. Peer review and data curation made the author clarify a few points and share all of the data and results in an open and well curated manner. The annotated genome of the lemon sole, with its high continuity, should therefore provide important reference data for future population genetic analyses and conservation strategies of this organism.

---

## [Reviewer Report]

Indicate in the comments box below whether you are happy with the changes made or if the manuscript is unacceptable.Comments on revised manuscriptThank you for addressing my comments. While I understand the study's limitations, including its focus as part of a university course and the use of a single specimen, I believe the manuscript lacks sufficient impact without exploring the genetic basis of sexual dimorphism or incorporating comparative analyses with other flatfish genomes. The genome assembly and annotation are well-executed, but the absence of biological context limits the broader relevance of the work. Sexual dimorphism in lemon sole, a commercially important species, is a key topic that could inform aquaculture and fisheries management. Without addressing this, the manuscript misses an opportunity to answer important scientific questions. For these reasons, I cannot recommend the manuscript for publication in its current form. While the technical work is solid, additional analyses or a broader scope are needed to enhance its contribution to the field

---

## [Reviewer Report]

Indicate in the comments box below whether you are happy with the changes made or if the manuscript is unacceptable.Comments on revised manuscriptThis study aims to perform chromosome-level genome assembly of the lemon sole (Microstomus kitt) and conduct a comprehensive analysis of its genome using high-throughput sequencing technology. Researchers utilized PacBio HiFi sequencing technology to carry out whole-genome sequencing of this species, resulting in a high-quality and complete genome sequence. The genome sequence has a length of 633 Mbp, with 23 chromosome-level sequences successfully assembled. Additionally, BUSCO analysis indicated that this genome sequence possesses a high level of completeness. These results suggest that the lemon sole genome sequence can serve as an important reference for future population genetic studies of commercially valuable edible fish species. However, there are certain issues with the paper that need to be addressed: The authors emphasize that female lemon soles grow larger than males, yet they chose to sequence the male genome instead of focusing on the more unique female. The authors should clarify this choice. The HI-C assisted assembly results show that male lemon soles have 23 chromosome pairs. Are there any heteromorphic chromosomes? The authors need to elucidate the karyotype of the lemon sole, as this information is significant for both the genome assembly and subsequent research. The survey results indicate a high level of heterozygosity in lemon sole. How did the authors account for this high heterozygosity to obtain a relatively complete genome? Could this affect the accuracy of the genome? Although the authors achieved high-quality genome results through PacBio sequencing, they used BUSCO for genome quality assessment. To further highlight the completeness and accuracy of the assembled genome, it is recommended that the authors utilize QV for additional evaluation. To ensure high levels of data sharing and reproducibility, the authors are requested to provide the chromosome-level genome fasta file and gff annotation file. In summary, the authors are encouraged to provide additional information and make necessary revisions.

---

## [Reviewer Report]

Reviewer name and names of any other individual's who aided in reviewer Alejandro MechalyDo you understand and agree to our policy of having open and named reviews, and having your review included with the published papers. (If no, please inform the editor that you cannot review this manuscript.)YesIs the language of sufficient quality?YesPlease add additional comments on language quality to clarify if needed
Are all data available and do they match the descriptions in the paper? NoAdditional CommentsThe BioProject number is not included in the submitted manuscript.Are the data and metadata consistent with relevant minimum information or reporting standards? See GigaDB checklists for examples <a href="http://gigadb.org/site/guide" target="_blank">http://gigadb.org/site/guide</a>NoAdditional CommentsThe BioProject number is not included in the submitted manuscript.Is the data acquisition clear, complete and methodologically sound?YesAdditional CommentsIs there sufficient detail in the methods and data-processing steps to allow reproduction?YesAdditional CommentsIs there sufficient data validation and statistical analyses of data quality? YesAdditional CommentsIs the validation suitable for this type of data?YesAdditional CommentsIs there sufficient information for others to reuse this dataset or integrate it with other data?NoAdditional CommentsAny Additional Overall Comments to the AuthorThe paper presents a valuable contribution to the genomics of Microstomus kitt (lemon sole), a commercially important species. The study introduces a chromosome-level genome assembly using PacBio HiFi sequencing, resulting in a highly contiguous assembly with 99.7% completeness in BUSCO genes. This high-quality genome will serve as a key resource for future population genomics and aquaculture studies. Overall, this assembly offers a solid foundation for advancing research on the biology and management of lemon sole. The main critique of this study is that, while it highlights the sexual dimorphism in lemon sole, where females are larger than males, it does not delve into this aspect in detail. Although the research presents valuable data through a high-quality chromosomal-level genome assembly, it focuses exclusively on male specimens. Comparing the genomes of both sexes would be highly insightful, potentially revealing the genetic mechanisms or pathways underlying this dimorphism through comparative genomics. Recent studies on flatfish (Villarreal et al., 2024. https://doi.org/10.1186/s12864-024-10081-z) have used comparative genomics to examine sex determination genes, and applying this approach to lemon sole would significantly enhance the study’s impact. Furthermore, there are numerous sequenced flatfish genomes that should be analyzed alongside these results to provide a more comprehensive context.RecommendationMajor Revision

---

## [Reviewer Report]

Reviewer name and names of any other individual's who aided in reviewer Yongshuang XiaoDo you understand and agree to our policy of having open and named reviews, and having your review included with the published papers. (If no, please inform the editor that you cannot review this manuscript.)YesIs the language of sufficient quality?YesPlease add additional comments on language quality to clarify if needed
Are all data available and do they match the descriptions in the paper? YesAdditional CommentsAre the data and metadata consistent with relevant minimum information or reporting standards? See GigaDB checklists for examples <a href="http://gigadb.org/site/guide" target="_blank">http://gigadb.org/site/guide</a>YesAdditional CommentsIs the data acquisition clear, complete and methodologically sound?YesAdditional CommentsIs there sufficient detail in the methods and data-processing steps to allow reproduction?NoAdditional CommentsIs there sufficient data validation and statistical analyses of data quality? NoAdditional CommentsIs the validation suitable for this type of data?YesAdditional CommentsIs there sufficient information for others to reuse this dataset or integrate it with other data?NoAdditional CommentsThis MS presents the chromosome-level genome assembly of Microstomus kitt, a species belonging to the Pleuronectidae family and mainly distributed in the North European seas. The study utilized PacBio HiFi sequencing technology combined with Hi-C data for chromosome-level assembly, resulting in a high-quality reference genome of approximately 633 MB, including 23 chromosomal length scaffolds, completing 99.7% of BUSCO genes, demonstrating high assembly completeness and gene annotation quality. Further analysis revealed abundant repetitive sequences and gene features in the lemon sole genome, providing important resources for future genetic studies of this species and its close relatives. The paper presents several issues as follows: 1. From the evaluation of the genome, the estimated size is around 542 Mb, while the manually curated Hi-C results yielded a genome size of 633 Mb. The authors are requested to explain why there is a difference of nearly 100 Mb between the second-generation sequencing evaluation and the third-generation results. 2. Utilizing PacBio HiFi sequencing technology, which generates long reads, and its associated assembly software, the authors were able to assemble the genome at the chromosome level. The authors explicitly state that the size of the 23 chromosomal level genomes assembled using YaHS and Chromap software is around 500 Mb, which is consistent with the genome survey results. How does the author know that the assembled genome is erroneous? 3. Based on the author's description, it is not clear what the size of the assembled genome from a single chain using PacBio sequencing is. The author needs to provide this data in the results. 4. The authors performed quality assessments of the assembled genome using various methods such as Merqury. However, the description of the evaluation results is lacking. The authors are requested to include the QV evaluation values and additional results of SNP alignment for the second-generation sequencing data. 5. For gene annotation, the authors used the genomes of five species of Pleuronectidae as references. We are eager to see the results of the alignment analysis between the genome obtained using PacBio Revio and the aforementioned five fish genomes. Although these results do not need to be included in the main text, they should be provided as part of the response to the reviewers, including the alignment results and alignment rates for both sets of assembled genomes (500 Mb and 633 Mb). 6. The authors are requested to include the length information of each chromosome in the supplementary files. From the assembly results, it appears that the PacBio Revio results are not as impressive as anticipated, particularly with a Scaffold N50 of 29.4 Mbp. Is this due to limitations in the length of the chromosomes themselves, affecting the quality metrics of this genome? 7. The data should be uploaded to NCBI and obtain the corresponding registration code.Any Additional Overall Comments to the AuthorThis MS presents the chromosome-level genome assembly of Microstomus kitt, a species belonging to the Pleuronectidae family and mainly distributed in the North European seas. The study utilized PacBio HiFi sequencing technology combined with Hi-C data for chromosome-level assembly, resulting in a high-quality reference genome of approximately 633 MB, including 23 chromosomal length scaffolds, completing 99.7% of BUSCO genes, demonstrating high assembly completeness and gene annotation quality. Further analysis revealed abundant repetitive sequences and gene features in the lemon sole genome, providing important resources for future genetic studies of this species and its close relatives. The paper presents several issues as follows: 1. From the evaluation of the genome, the estimated size is around 542 Mb, while the manually curated Hi-C results yielded a genome size of 633 Mb. The authors are requested to explain why there is a difference of nearly 100 Mb between the second-generation sequencing evaluation and the third-generation results. 2. Utilizing PacBio HiFi sequencing technology, which generates long reads, and its associated assembly software, the authors were able to assemble the genome at the chromosome level. The authors explicitly state that the size of the 23 chromosomal level genomes assembled using YaHS and Chromap software is around 500 Mb, which is consistent with the genome survey results. How does the author know that the assembled genome is erroneous? 3. Based on the author's description, it is not clear what the size of the assembled genome from a single chain using PacBio sequencing is. The author needs to provide this data in the results. 4. The authors performed quality assessments of the assembled genome using various methods such as Merqury. However, the description of the evaluation results is lacking. The authors are requested to include the QV evaluation values and additional results of SNP alignment for the second-generation sequencing data. 5. For gene annotation, the authors used the genomes of five species of Pleuronectidae as references. We are eager to see the results of the alignment analysis between the genome obtained using PacBio Revio and the aforementioned five fish genomes. Although these results do not need to be included in the main text, they should be provided as part of the response to the reviewers, including the alignment results and alignment rates for both sets of assembled genomes (500 Mb and 633 Mb). 6. The authors are requested to include the length information of each chromosome in the supplementary files. From the assembly results, it appears that the PacBio Revio results are not as impressive as anticipated, particularly with a Scaffold N50 of 29.4 Mbp. Is this due to limitations in the length of the chromosomes themselves, affecting the quality metrics of this genome? 7. The data should be uploaded to NCBI and obtain the corresponding registration code.RecommendationMajor Revision